# Immunotherapy of Glioblastoma: Current Strategies and Challenges in Tumor Model Development

**DOI:** 10.3390/cells10020265

**Published:** 2021-01-29

**Authors:** Bernarda Majc, Metka Novak, Nataša Kopitar-Jerala, Anahid Jewett, Barbara Breznik

**Affiliations:** 1Department of Genetic Toxicology and Cancer Biology, National Institute of Biology, 111 Večna pot, SI-1000 Ljubljana, Slovenia; bernarda.majc@nib.si (B.M.); metka.novak@nib.si (M.N.); 2International Postgraduate School Jozef Stefan, 39 Jamova ulica, SI-1000 Ljubljana, Slovenia; 3Department of Biochemistry, Molecular and Structural Biology, Jozef Stefan Institute, 39 Jamova ulica, SI-1000 Ljubljana, Slovenia; natasa.kopitar@ijs.si; 4Division of Oral Biology and Medicine, The Jane and Jerry Weintraub Center for Reconstructive Biotechnology, University of California School of Dentistry, 10833 Le Conte Ave, Los Angeles, CA 90095, USA; ajewett@dentistry.ucla.edu

**Keywords:** glioblastoma, immunotherapy, tumor model, stem cell, organoid, heterogeneity, immunosuppression, microenvironment

## Abstract

Glioblastoma is the most common brain malignant tumor in the adult population, and immunotherapy is playing an increasingly central role in the treatment of many cancers. Nevertheless, the search for effective immunotherapeutic approaches for glioblastoma patients continues. The goal of immunotherapy is to promote tumor eradication, boost the patient’s innate and adaptive immune responses, and overcome tumor immune resistance. A range of new, promising immunotherapeutic strategies has been applied for glioblastoma, including vaccines, oncolytic viruses, immune checkpoint inhibitors, and adoptive cell transfer. However, the main challenges of immunotherapy for glioblastoma are the intracranial location and heterogeneity of the tumor as well as the unique, immunosuppressive tumor microenvironment. Owing to the lack of appropriate tumor models, there are discrepancies in the efficiency of various immunotherapeutic strategies between preclinical studies (with in vitro and animal models) on the one hand and clinical studies (on humans) on the other hand. In this review, we summarize the glioblastoma characteristics that drive tolerance to immunotherapy, the currently used immunotherapeutic approaches against glioblastoma, and the most suitable tumor models to mimic conditions in glioblastoma patients. These models are improving and can more precisely predict patients’ responses to immunotherapeutic treatments, either alone or in combination with standard treatment.

## 1. Introduction: Glioblastoma and Its Heterogeneity

The most aggressive and also most common primary brain tumor in adults is glioblastoma (Glioblastoma WHO grade IV). Glioblastoma is poorly responsive to therapy, which includes maximal surgical removal that is followed by chemotherapy and radiation therapy and has one of the shortest survival rates amongst all cancers [1]. For example, tumor treating fields treatment together with chemotherapy improved median overall survival of glioblastoma patients from 16 to 20.9 months [2]. Despite novel modalities in treatment, which rely on the Stupp protocol from 2005, the 5-year survival rate of patients is less than 5% [3,4,5]. Glioblastoma has distinct histological characteristics, including a pleomorphic cell composition, increased mitotic and cellular activity, and significant angiogenesis and necrosis [6]. The poor response of glioblastoma to treatment and its poor prognosis are associated with diffused invasion patterns within the central nervous system (CNS) [7]. Furthermore, the blood-brain barrier (BBB) presents both a physical and biochemical barrier to the CNS for large molecules [8,9]. Lymphatic vessels have been found in the meninges of humans and mice [10,11,12], causing the notion of the CNS as an immune-privileged system to be reconsidered. Brain-resident macrophages, i.e., microglia, are also now broadly recognized as antigen-presenting cells of the CNS. Although the brain is an immunologically distinct site, the brain microenvironment is capable of generating robust immune responses and offers adequate opportunities for the implementation of brain tumor immunotherapy [13]. In addition, the BBB can be disrupted in brain tumor patients, which increases the infiltration of immune cells into the tumor area. However, most GBM patients have variable regions of disrupted BBB, meaning that tumor regions with disrupted BBB and tumor regions with intact BBB exist [14].

The successful treatment of glioblastoma remains one of the most difficult challenges in brain cancer therapy. This is due to (1) the small population of therapy-resistant glioblastoma stem cells (GSCs) [15,16,17,18] and (2) inter- and intra-tumor heterogeneity that consists of a variety of different subtypes of glioblastoma [19] and stromal cells in the tumor microenvironment (TME) [20,21]. Glioblastomas have been genetically categorized by The Cancer Genome Atlas into three subtypes: proneural, classical, and mesenchymal. Each of these subtypes is characterized by mutations causing platelet-derived growth factor receptor alpha activation, epidermal growth factor receptor (EGFR) activation, and neurofibromin 1 deletions, respectively. Glioblastoma subtypes differ in their prognostic value, with mesenchymal and proneural subtypes exhibiting the shortest and longest overall survival rates, respectively [19]. Moreover, the composition of the TME is linked to the molecular subtypes of glioblastoma. Mesenchymal tumors contain abundant gene expression signatures for macrophages, CD4^+^ T cells, and neutrophils [22]; this is also associated with a higher glioma grade [19]. An increase in macrophages and microglia cells occurs upon disease recurrence and is associated with shorter relapse time after therapy [22].

GSCs are largely responsible for glioblastoma recurrence and therapy resistance due to their DNA repair and multi-drug resistance mechanisms as well as their ability to evade the immune response [15,23,24]. GSCs are maintained in hypoxic and peri-arteriolar GSC niches [25,26] and are more abundant in more aggressive, high-grade tumors with worse prognoses [27,28]. The glioblastoma TME regulates and determines the cellular state and drives GSC plasticity [29], which leads to the therapeutic resistance of tumors [30].

The predominant immune cells in the brain are macrophages, more specifically, tissue-resident macrophages known as microglia [31]. In brain cancer or other brain inflammatory conditions, additional peripheral monocytes are recruited from bone marrow and are differentiated in the brain into macrophages that are phenotypically distinct from microglia [32,33]. Immune cells are recruited and phenotypically changed by glioblastoma cells; this supports tumor growth and an immunosuppressive TME [34] through the release of cytokines, extracellular vesicles, and connecting nanotubes [35]. Chemoattraction between cells is mediated by members of a large family of chemokines [36,37]. For example, in glioblastoma, the chemokine (C-C motif) ligand 5 (CCL5) and its receptor C-C chemokine receptor type 5 (CCR5) are involved in autocrine and paracrine cross-talk between glioblastoma cells and the TME, contributing to stromal and immune cell tumor infiltration and glioblastoma cell invasion [38,39]. The attraction between endothelial and glioblastoma cells in GSC niches is predominantly maintained by the binding of C-X-C motif chemokine 12 (CXCL12, also known as stromal cell-derived factor 1α (SDF-1α)) to the C-X-C chemokine receptor type 4 (CXCR4) in GSCs [26].

## 2. The Immunosuppressive Microenvironment of Glioblastoma

Multi-layered immunosuppression exists in glioblastoma, both at the systemic and local level [40]. Systemic immunosuppression in glioblastoma patients is, to a large extent, induced by standard treatment including radiotherapy, temozolomide, and corticosteroids, which weakens the adaptive and innate immune responses [41]. Moreover, defects in antitumor responses arise from defective T cell mobilization from the periphery due to T cell entrapment in the bone marrow, which is caused by the loss of the surface sphingosine-1-phosphate receptor 1 (S1P1) [42,43] that binds the lipid second messenger sphingosine-1-phosphate (S1P) [44]. The S1P-S1P1 axis plays a role in governing lymphocyte trafficking. Naïve T cell egress from bone marrow or secondary lymphoid organs cannot occur without functional S1P1 on the cell surface, as S1P1 is essential for lymphocyte recirculation [42,45].

The glioblastoma microenvironment is extremely immunosuppressive due to its low immunogenicity, the immunosuppressive properties of many cells (including cancer cells, cancer stem cells (CSCs), and tumor-infiltrating immunosuppressive immune cells, e.g., myeloid cells and T regulatory cells (Tregs)), and the lack of antigen-presenting potential and costimulatory antigens, leading to tumor resistance to immunotherapy.

Glioblastoma cells and GSCs employ several mechanisms to evade the immune response. These include their intrinsic resistance to the induction of cell death, modulation of tumor antigens and cell surface molecules (which are important for the recognition and destruction of immune effector and antigen-presenting cells), and secretion of extracellular vehicles, cytokines, and growth factors. For example, glioblastoma cells express the programmed cell death receptor 1 ligand (PD-L1) that inhibits the cytotoxicity of cytotoxic T cells and downregulates major histocompatibility complex (MHC) class I, resulting in deficient T cell cytotoxicity [40]. Moreover, glioblastoma cells may increase the expression of natural killer (NK) cell inhibitory ligands and decrease the expression of NK cell-activating NK group 2 member D (NKG2D) ligands, leading to inhibited NK cell-mediated lysis [46].

Glioblastoma is immunologically a cold tumor with low NK and T cell infiltration compared to other solid tumors. In glioblastoma, T and NK cells become dysfunctional. T cells are senescent, tolerant, exhausted, and anergic due to the immunosuppressive glioblastoma TME [40,47]. NK cells are important as immune effectors of the first line of defense against tumor cells and have been shown to control metastasis by eliminating circulating cancer cells [48]. The proposed mechanisms for the functional inactivation of tumor-associated NK cells are the overexpression of Fas ligand, the loss of mRNA for granzyme B [49], and the decrease of CD16 and its associated zeta chain [50,51,52]. T and NK cell dysfunction is also caused by co-expression of multiple co-inhibitory receptors, including programmed cell death protein 1 (PD-1), T cell immunoglobulin and mucin-domain containing-3 (TIM3), lymphocyte activation gene 3 protein (LAG3), cytotoxic T lymphocyte-associated protein 4 (CTLA-4), and T cell immunoreceptor with immunoglobulin and ITIM domains (TIGIT) [53].

Glioblastoma immunosuppressive TME is driven by tumor-intrinsic factors and brain (host) tissue responses to tumor antigens, such as overexpression of the indoleamine 2,3-dioxygenase (IDO) enzyme [54,55] and oncogene transforming growth factor-beta (TGF-β), respectively. IDO is a tryptophan catabolic enzyme overexpressed in several tumor types that creates an immunosuppressive microenvironment via the suppression of cytotoxic (CD8^+^) T cell proliferation and effector function [56] and the promotion of Treg generation via an aryl hydrocarbon receptor-dependent mechanism [56]. Cytokines, such as IL-10 and TGF-β, within the glioblastoma TME cause microglia to lose MHC expression [57,58]. TGF-β reduces NK and CD8^+^ T cell activation through inhibiting NKG2D expression, which is responsible for inducing lysis of NKG2D ligand-bearing cells that express class I MHC-related proteins, MHC Class I Polypeptide-Related Sequence A (MICA) and B, and the UL16 binding protein (ULB) 1–4 protein family [59].

Glioblastoma cells in the TME hijack many different cells to support tumor growth through the recruitment and suppression of many cells of the innate and adaptive immune responses [20]. For example, Tregs and myeloid-derived suppressive cells that inhibit the proliferation and activation of effector cells (i.e., T cells and NK cells) and antigen-presenting cells are recruited. Increased numbers of forkhead box P3 (FOXP3)^+^ Tregs were found in glioblastoma [60,61]; however, their correlation with patient survival was modest [60,62,63]. Microglia and tumor-infiltrating macrophages influence immunosuppression by secreting the cytokine IL-10, TGF-β, and extracellular vesicles [64,65]. These complex interactions open new therapeutic windows for glioblastoma treatment. Colony-stimulating factor 1 (CSF-1) is a potent chemoattractant that regulates the differentiation of monocytes into tumor-associated macrophages (TAMs), and its overexpression correlates with increased TAM infiltration and poor clinical outcomes [66]. Inhibition of the CSF-1 receptor (CSF-1R) enhanced sensitivity to irradiation by altering both the recruitment and the phenotype of myeloid-derived cells recruited to the irradiated glioblastoma [67]. TAMs also express high levels of PD-L1 [41]. Moreover, hypoxic conditions in the glioblastoma TME, through increased hypoxia-inducible factor (HIF) transcription factors and vascular endothelial growth factor (VEGF), increase TAM tumor infiltration [40].

## 3. Immunotherapeutic Strategies for Glioblastoma

The goal of immunotherapy is to stimulate patient antitumor immunity and eliminate glioblastoma cells, specifically the therapy-resistant fraction of glioblastoma cells. Several immunotherapeutic approaches, including vaccines, oncolytic viruses, checkpoint inhibitors, and adoptive cellular transfer (chimeric antigen receptor (CAR) T and NK cells), alone or in combination with standard glioblastoma therapy, have been tested against glioblastoma in preclinical and clinical studies [13,41,68,69,70,71].

### 3.1. Vaccines

The main goal of the vaccine-based approach is to strengthen the adoptive immune response in the brain against glioblastoma cells. Several vaccines with peptides, mimicking neoantigens in glioblastoma cells, have been developed to trigger an antitumor immune response in patients. Vaccination of glioblastoma patients with a peptide mimicking the EGFR variant III (EGFRvIII) in glioblastoma cells, together with standard temozolomide chemotherapy or the anti-angiogenic agent bevacizumab, showed promising anti-glioblastoma effects in clinical trials. As only 25–30% of patients express EGFRvIII, and its expression is heterogeneous in tumors and unstable through the course of the disease, the efficiency of these vaccines is limited [72,73]. Moreover, a randomized, double-blind, and international phase 3 trial, which assessed the efficacy of the vaccine, based on EGFRvIII-specific peptide (CDX-110), with temozolomide did not show a survival benefit for newly diagnosed glioblastoma patients with EGFRvIII mutation [74]. To overcome glioblastoma cell heterogeneity, multi-peptide vaccines based on the administration of a combination of tumor-associated peptides overexpressed in glioblastoma cells were developed; however, the overall survival of glioblastoma patients was not significantly improved [41,75]. The advantages of dendritic cell-based therapies are the induction of antitumor T cell responses and enhancement of tumor immunogenicity due to their antigen-presenting functions and ability to link innate immunity with adoptive immunity. This is extremely important, especially in low immunological tumors such as glioblastoma. Vaccines based on autologous dendritic cells, which can be primed ex vivo using patient-derived tumor lysates, CSCs, or glioblastoma-associated antigens, have been tested in several clinical trials together with temozolomide as standard treatment [69,76,77]. Based on those findings, vaccination induces immune responses, even antitumor T cell responses have been observed; however, immune stimulation seems to be insufficient to translate into clinical benefit, and thus the efficacy of vaccine immunotherapy is limited [41,77,78]. Recent clinical studies are utilizing personalized vaccines that target a patient’s unique tumor-associated neoantigens [41].

### 3.2. Oncolytic Viruses

Virus-based anticancer therapies are based on viruses that selectively infect or replicate in tumor cells, leading to the lysis of infected tumor cells (direct effect) and the activation of immunogenic tumor cell death pathways that can stimulate antigen presentation and the adaptive antitumor immune response (indirect effects). Additionally, oncolytic viruses activate the innate immune system through pattern recognition receptors and pathogen-associated molecular patterns [79]. Current oncolytic viral approaches utilize replication-competent viruses, such as retroviruses, adenoviruses, herpes simplex viruses, polioviruses, and measles viruses [13,41]. Such viral approaches also include oncolytic viruses that are armed with immunoregulatory inserts, such as interleukin 12 and OX40 ligand, further boosting innate and adoptive antitumor immune responses [70,80]. Adenoviruses can be modified to become tumoricidal gene delivery vectors, such as the adenoviral vector AdV-tk. This vector contains the herpes simplex virus thymidine kinase gene, which converts the toxic nucleotide analog, the prodrug ganciclovir or valacyclovir that kill fast-growing tumor cells. Moreover, induced cell death of tumor cells elicits immune effects. In phase II of clinical trials for newly diagnosed malignant gliomas, local delivery of AdV-tk plus valacyclovir together with standard treatment improved progression-free and overall survival by a few months [81]. A non-lytic, replicating retrovirus encoding cytosine deaminase has been used in clinical trials in combination with the prodrug 5-fluorocytosine, which is converted in virus-infected tumor cells into the antimetabolite 5-fluorouracil by exogenous cytosine deaminase, which is not otherwise expressed in human cells. This combined viral treatment prolonged the survival of patients with primary and recurrent high-grade gliomas in phase I clinical trials, increased immunogenicity within the TME, and activated the adoptive immune response [41,82]. Oncolytic viral immunotherapy can sensitize cancer patients to other active immunotherapeutic approaches; however, the marginal increases in overall survival have not yet achieved clinical translation. Namely, viruses and viral vectors show low transfection rates and limited penetration of brain tumors [83]. The combined approach with other immunotherapies, including immune checkpoint inhibitors and adoptive cell therapy, is currently the main focus aiming to prolong oncolytic virus-initiated clinical responses [79,84].

### 3.3. Immune Checkpoint Inhibitors

Immune checkpoint inhibitors are antibodies, which reduce the activity of endogenous negative regulatory pathways that limit T cell activation. Antibodies that block the inhibitory immune checkpoint proteins CTLA-4, PD-1, and its ligand PD-L1 have shown major improvements in the outcome of cancer patients in the past decade and are widely used. CTLA-4 and PD-1 are expressed on T cells, whereas PD-L1 is expressed on certain subsets of immune cells, including TAMs, and is aberrantly expressed on tumor cells. PD-L1 expression has been found in glioblastoma cells; however, not all glioblastomas express PD-L1 and its expression changes during the course of the disease [85]. Although there were several encouraging preclinical data on the use of immune checkpoint inhibitors (anti-PD-1 and anti-CTLA-4 antibodies, alone or in combination) for glioblastoma, clinical trials have been disappointing, with no patient survival improvement [41,85]. Several reasons for the poor efficacy of immune checkpoint inhibitors in glioblastoma have been identified, including the timing of delivery (neoadjuvant or adjuvant therapy), BBB, low infiltration of T cells into the tumor, predominant myeloid infiltrate, and multi-layered immunosuppression in the TME [84,85,86,87,88]. A subgroup of glioblastoma patients have benefited from immune checkpoint inhibitor treatment and have exhibited prolonged survival. The tumors of these patients have enriched alterations in the mitogen-activated protein kinase (MAPK) pathway (mutationally activated protein tyrosine phosphatase non-receptor type 11 (PTP11) and B-raf murine sarcoma (BRAF)) [41]. In the same study, non-responders to immune-checkpoint inhibitors exhibited phosphatase and tensin homolog (PTEN) mutations that were associated with immunosuppressive expression signatures [41]. A recent study by Cloughesy et al. have shown that patients with recurrent glioblastoma received neoadjuvant treatment with pembrolizumab (anti-PD-1), with continued adjuvant therapy following surgery, had significantly improved overall survival compared to that receiving only adjuvant post-surgical treatment with pembrolizumab. Neoadjuvant administration of pembrolizumab enhanced local and systemic immune responses in patients [89]. Currently, clinical trials with combinatorial therapy, in which immune checkpoint inhibition is combined with other immunostimulatory approaches, are in progress [84,85,86,87,88].

### 3.4. Adoptive Cell Therapies: CAR T and NK Therapy

Genetically modified T cells that express CARs consist of an extracellular tumor-specific antigen-recognition domain and a T cell activation domain. A great advantage of CAR T cells is that they can recognize specific antigens and trigger cell lysis independently of major MHC I presentation. After autologous or allogeneic T cells are engineered in the laboratory, they are adoptively transferred into the patient to activate the antitumor immune response. In the case of brain tumors, CAR T cells can be applied intravenously, intracranially, or into the tumor [90]. CAR T cells can target glioblastoma-specific antigens, including interleukin-13 receptor subunit alpha-2 (IL-13Rα2), EGFR wt, and EGFRvIII, and are thus effective against glioblastoma in preclinical models [13,91,92]. In addition, glioblastoma patients who received IL-13Rα2- and EGFRvIII-targeting CAR T cells showed clinical responses in early clinical studies. CAR T cells can infiltrate the glioblastoma, become activated within the glioblastoma microenvironment, and activate various adoptive cell responses in patients. However, CAR T cells must be combined with other therapies or with CAR T cells targeting multiple different antigens because of glioblastoma heterogeneity, tumor antigen loss during tumor progression, CAR T exhaustion in the TME, activation of compensatory adoptive resistance mechanisms, and upregulation of immunosuppressive factors and cells (e.g., IDO1, PD-L1, and Tregs) in the TME that are triggered after CAR T cell application. Trivalent CART T cells co-targeting human epidermal growth factor receptor 2 (HER2), IL-13Rα2, and EPH receptor A2 (EphA2) have been demonstrated to be more efficacious in preclinical studies than bivalent or monovalent CAR T cells [13,90,93]. CAR T cells targeting tumor-initiating cells through the surface receptor CD133 in glioblastoma have been developed recently. CD133 (prominin 1) has been identified as a surface biomarker of tumor-initiating and therapy-resistant GSCs [94]. Intracranial injection of CD133-specific CAR T cells reduced tumor burden and prolonged survival of glioblastoma-bearing mice. This treatment is considered safe in mice, as it did not incur acute toxicity in normal hematopoietic stem and progenitor cells that also express CD133 [95].

NK cells are the only immune effectors known to recognize and kill GSCs without requiring approaches that generate immunogenic antigens and enable cell priming with appropriate costimulatory signals, as are required for potential T or dendritic cell-based immunotherapies. NK cells preferentially recognize and lyse GSCs in a non-MHC restricted manner [96]. NK cells are the main mediators of antibody-dependent cellular cytotoxicity [46,68]. The use of allogeneic NK cells is preferred because the inhibitory killer-cell immunoglobulin-like receptors (KIR) receptors on the surface of donor NK cells cannot recognize self-MHC class I molecules on the tumor cells of the patient. Consequently, the absence of inhibitory signals allows NK cell activation [46,96]. As NK cells have been shown to preferentially kill GSCs [97,98] and penetrate the BBB [99] in preclinical in vitro and animal models when administered systematically, patients with glioblastoma and high-grade gliomas are now undergoing allogeneic and autologous NK cell administration in clinical trials or are undergoing recruitment (NCT04489420, NCT04254419: ClinicalTrials.gov). To increase natural NK cytotoxicity and attack towards tumors with a heterogeneous expression of CAR target antigens, NK cells can be genetically engineered to express CARs. CAR NK cells targeting the glioblastoma cell-specific antigens EGFR, EGFRvIII, and HER2 have been generated from NK cells derived from the following: the peripheral blood of healthy donors, umbilical cord blood, induced pluripotent stem cells, and the NK-92 cell line, which all display features of activated primary NK cells. CAR NK cells exhibited GSC and differentiated glioblastoma cell cytotoxicity increased levels of interferon-gamma (IFN-γ), and prolonged survival of glioblastoma-bearing mice in preclinical studies [68,100,101]. Currently, glioblastoma patients are being recruited for clinical trials using HER-2-specific CAR NK cells (NCT03383978: ClinicalTrials.gov).

### 3.5. Resistance to Immunotherapy and Combinatorial Approaches

As single immunotherapeutic approaches have shown some promising results but are not sufficiently successful in prolonging the survival of glioblastoma patients, combinatorial immunotherapeutic approaches that can synergize together are now under investigation. The reasons for the poor response to single immunotherapeutic approaches are adoptive tumor resistance compensatory mechanisms due to multi-layered immunosuppression, local immune cell dysfunction, and glioblastoma tumor heterogeneity [91]. Specific efforts to facilitate the antitumor immune response are focused on targeting the immunosuppressive myeloid compartment, reducing the activity of immunosuppressive molecules (e.g., IDO and CSF-1R), and activating antitumor functions of other immune cells, NK cells, and dendritic cells [41]. Anti-IDO in combination with anti-PD-1 and anti-CTLA-4 approaches are more potent than monotherapy and decrease the accumulation of Tregs in a glioblastoma murine model [86]. The synergistic effects of combining adenovirus-based therapy and anti-PD-1 result in prolonged survival in experimental models of glioblastoma [13,80]. Although CSF-1R inhibitors showed promising results in preclinical studies, the clinical trials with orally administered CSF-1R inhibitor PLX-3397 were negative, with minimal clinical efficacy in patients with recurrent glioblastoma. Microenvironment-driven resistance to CSF-1R inhibitor is mediated through phosphatidylinositol 3-kinase (PI3K) pathway, which was elevated and driven by insulin-like growth factor–1 (IGF-1) and tumor cell IGF-1 receptor (IGF-1R) [102,103]. The use of anti-CSF-1R agents with anti-PD-1 therapy is now in clinical trials [13]. Moreover, CAR T therapy (anti-HER2, anti-IL-13Rα2, and anti-EGFRvIII) in combination with CTLA-4 or PD-1 inhibition has improved the effects in preclinical models and is now in clinical trials [90].

Current standard-of-care treatment for glioblastoma includes maximal surgical tumor resection, hyperfractionated radiotherapy, and temozolomide, which, in combination with commonly used corticosteroids, systemically weakens the immune system, increases immunosuppression, and hinders the immunotherapeutic strategy [41]. It has also been shown that a standard dose of temozolomide induces immunosuppression and abrogates the effect of anti-PD-1 therapy [104] and oncolytic virus-based immunotherapy [105]. Conversely, localized treatment, which increases the availability of tumor antigens, synergizes with immunotherapy. It has been shown that radiation increases the mutational burden of tumors and triggers tumor necrosis and antigen release, leading to increased antigen presentation and immunogenicity [106]. The high mutational burden is associated with response to immunotherapy in several types of cancer, but not in gliomas. For example, gliomas with a high mutational burden and mismatch repair gene deficiency are less responsive to PD-1 blockage [107]. Low mutation burden in recurrent glioblastoma patients was recently associated with longer survival after immunotherapy, implicating that tumor mutational burden itself may not be a causative driver of response to immunotherapy, but may reflect the immunological status of tumor or some other co-related feature, among them time to recurrence, TP53 mutation and any differences in the clinical care between patients with high vs. low mutational burden [108]. The combination of immunotherapy with hypofractionated stereotactic radiosurgery can probably improve the efficacy of immunotherapy as stereotactic radiosurgery does not trigger systemic immunosuppression [13]. Metronomic dosing of temozolomide or local chemotherapy are preferred when combining temozolomide with immunotherapy [104]. However, additional studies are needed to elucidate the efficiency of these combinatorial approaches.

## 4. Advanced In Vitro and Animal Tumor Models for Testing Immunotherapeutic Approaches

Glioblastomas are very heterogeneous in their cellular composition, gene expression, and phenotypic properties [109]. In addition, glioblastoma contains a unique and complex immune TME. Based on studies on preclinical tumor models and clinical stages, we conclude that the currently used glioblastoma tumor models do not sufficiently reflect the conditions in humans, as several immunotherapeutic strategies that were efficient in preclinical studies failed to demonstrate sufficient clinical significance. The ability to comprehensively understand glioblastoma phenotypes and mimic their specific therapeutic responses to enable personalized therapy requires the creation of clinically relevant models that reliably reflect the complexity of the tumor in humans. For example, current patient-derived tumor models lack clinically relevant recapitulation of immune compartments [110]. To address all these challenges, different tumor models have been developed, including CSCs, organoids, patient-derived xenografts, genetically engineered mice models, and humanized mice. Comparisons of various tumor models to explore immunotherapeutic approaches and their advantages and disadvantages are listed in Table 1.

### 4.1. CSCs

Considering the importance of targeting therapy-resistant and tumorigenic CSCs, 3D models of CSCs incorporate the cellular heterogeneity of tumors, improve drug response predictability, and represent better models for discovering new targets for anticancer drugs compared to traditional 2D tumor cell lines [125]. GSC tumorspheres represent models generated by the symmetric and asymmetric division of patient-derived GSCs in a defined medium supplemented with growth factors, i.e., epidermal growth factor (EGF), fibroblast growth factor (FGF)-2, and neuronal viability supplement B27 [112]. These factors and the absence of serum are needed to maintain self-renewal and proliferation and to preserve the genetic characteristics observed in patients’ samples. Tumorspheres are characterized by an external proliferating zone, intermediate quiescent zone, and an inner necrotic core [126], observed at a certain distance from the presence of nutrients, metabolites, and oxygen, resembling the necrotic areas of in vivo glioblastoma [127]. Tumor cells within tumorspheres closely interact with each other, thus reproducing the physical communication and signaling pathways that affect proliferation, survival, and response to therapy in vivo [128] and forming a physical barrier that prevents and limits the transport of drugs into the tumorsphere mass [129]. Although a better model than monolayer cultures, tumorspheres represent random aggregations of cells that do not organize into tissue-like structures and also lack extracellular matrix [130]. The greater limitation of these models is the lack of neighboring non-tumor cells, i.e., stromal cells, including astrocytes, neurons, endothelial cells, mesenchymal stem cells, brain-resident microglia, and infiltrated peripheral immune cells; this altogether prevents studying their interactions with GSCs in vitro. Tumorspheres can be optimized by co-culturing cancer and stromal cells in so-called heterotypic spheroids, especially for testing cancer immunotherapeutic agents. For example, GSC tumorspheres were used to evaluate the penetration and cytotoxicity of highly cytotoxic super-charged NK cells (Figure 1, our results), grown in the presence of osteoclasts and probiotic bacteria to stimulate their cytotoxic potential towards CSCs [113]. In the study of Cheema et al. [131], the authors used a murine GSC model in syngeneic immunocompetent mice to test a genetically engineered oncolytic herpes simplex virus that is armed with the cytokine interleukin 12 (G47∆-mIL12). In addition to targeting GSCs, oncolytic virus treatment increased IFN-γ release, inhibited angiogenesis, and reduced the number of Tregs in the tumor.

### 4.2. Organotypic Tissue Slices

Organotypic tissue slice model of glioblastoma represents precision-cut slices of tumor tissue, in which the original inter and intra-tumor heterogeneity and the architecture of the tumor are maintained. Slices of the tumor are prepared with an automated vibratome and transferred onto membrane culture inserts for mechanical support in a specific cultivation medium [114]. This technique is relatively fast, it does not involve selective outgrowth of tumor cells, and therefore can be used for personalized treatment. Organotypic cultures have been used to study the invasive properties of glioblastoma and the patient-specific effect of anti-invasive drugs [115]. Recently an organotypic slice culture technique was developed from fresh pancreatic ductal adenocarcinoma to study the immune response after immunotherapy treatment [132] and can be applied to a variety of solid tumors, including glioblastoma. A disadvantage of this model is its relatively low throughput. The technique is laborious and requires specialized analysis tools.

### 4.3. Organoids

Organoids are 3D constructs composed of multiple cell types with the ability to self-organize and recapitulate the architecture and functionality of the original organ [110,133]. Different approaches for organoid generation have been applied, including using patient-derived adult stem cells and resected tumor tissues, as first described by Sato et al. [134]. Another approach involves the use of pluripotent stem cells, i.e., pluripotent embryonic stem cells and induced pluripotent stem cells [135]. The term “organotypic tumor spheroid” was initially used at the beginning of organoid development but was later replaced by the term “tumor organoid” [110]. Compared with traditional models, different tumor organoids, including liver [136], pancreatic [137], gastric [138,139], bladder [140], breast [141], and ovarian [142], show a vast potential for basic cancer research, drug screening, and personalized medicine and may bridge the gap between in vitro and in vivo cancer models. Until recently, it was unclear whether various methods for organoid preparation can be adapted for organoids from non-epithelial tumors. In 2016, Hubert et al. generated patient-derived glioblastoma organoids to study the heterogeneity and hypoxic gradient of tumors using a submerged culture system [116]. In this protocol, finely minced tumor specimens are embedded in a solid gel of extracellular matrix (Matrigel) to form 3–4 mm large organoids in the tissue culture medium, supplemented with EGF, FGF, and B27. These organoids formed in 2 months and could be cultured for over a year. Glioblastoma organoids are characterized by rapidly proliferating cells on the edge of the organoid and highly resistant quiescent CSCs in the hypoxic core with different molecular profiles. Although this is a very promising model of glioblastoma that closely resembles tumor sensitivity *in vitro*, its genetic and molecular features remain unclear. In 2018, Ogawa et al. constructed cerebral organoids using induced pluripotent stem cells and embryonic stem cells and induced glioma carcinogenesis by CRISPR/Cas9 technology to disrupt the TP53 tumor suppressor and express oncogenic HRas^G12V^ [110]. Moreover, neoplastic cerebral organoids were established by Bian et al. [118] via recapitulating brain tumorigenesis by introducing oncogenic mutations or amplifications in cerebral organoids using transposon-mediated gene insertion and CRISPR/Cas9 technology. These organoids developed *CDKN2A^−/−^*/*CDKN2B^−/−^*/*EGFR^OE^*/*EGFRvIII^OE^*, *NF1^−/−^/PTEN^−/−^/TP53^−/−^,* and *EGFRvIII^OE^/CDKN2A^−/−^/PTEN^−/−^* genotypes, which are commonly found in glioblastoma. In contrast to the aforementioned technique, induced pluripotent stem cells and embryonic stem cell organoids represent 3D human tissues generated by directed differentiation, self-morphogenesis, and intrinsically driven self-assembly of cells, recapitulating human organogenesis in vitro [143]. This type of organoid can contain multiple tissue cell types, including stroma and vasculature, unlike organoids developed from tissue-specific stem cells [144]. A novel approach using hESC-derived cerebral organoids and patient-derived GSCs to model tumor cell invasion was recently developed, i.e., a glioma cerebral organoid model. This system was shown to recapitulate the cellular behavior of glioblastoma and to maintain genetic aberrations found in the original tumor [145]. In a very recent study, Jacob et al. [117,146] established patient-derived glioblastoma organoids that accurately recapitulate the molecular, genetic, and cell-type heterogeneity of parental tumors. Compared to other previous protocols of glioblastoma organoids [116,118,119], the authors dissected tumor tissues into approximately 1 mm fragments without the addition of extracellular matrix or EGF and bFGF and cultured them on an orbital shaker for 1–2 weeks to generate 3D structures. These organoids contain heterogeneous populations of cellular subtypes and recapitulate tumor cell phenotypes, as confirmed by histopathology, single-cell RNA sequencing, and molecular profiling analysis. Moreover, glioblastoma organoids develop a hypoxic gradient and retain vasculature and TME composition, which mimics the main features of glioblastoma [117].

Organoids are becoming a very useful platform for cancer research, especially in the field of immuno-oncology; however, organoid establishment and its (pre)clinical applications are still immature. To date, co-cultures of epithelial tumor organoids and additional cellular components have been used to include the interactions between tumor and immune cells and have thus established a better preclinical model for immunotherapy. Immunocompetent organoids can be achieved by adding pre-treated autologous or allogeneic peripheral blood mononuclear cells (PBMCs) or specific immune cell populations, such as TAMs and tumor-infiltrating lymphocytes [110]. For example, in a recent study, Dijkstra et al. [147] enriched tumor-reactive T cells by co-culturing PBMCs and tumor organoids from colorectal and non-small-cell lung cancer and demonstrated that these T cells can be used to assess the efficiency of killing tumor organoids. In another study, gamma delta 2 (γδ2)^+^ T cells were co-cultured with organoids from human breast epithelia, and these lymphocytes effectively eliminated triple-negative breast cancer cells [148]. These and other studies demonstrate that T cells can be obtained and activated by organoids for adoptive T cell therapy. Using the air-liquid interface technique, Neal et al. [149] generated patient-derived organoids from different surgically resected primary and metastatic tumors with native embedded immune cells (CD8^+^ and CD4^+^ T cells, B cells, NK cells, and macrophages). This demonstrated the potential of organoids as tools to predict clinical responses to immune checkpoint therapies. For this method, tumor tissue fragments are embedded in a type I collagen matrix on an inner Transwell insert. Culture medium with different supplements is added to the outer dish to diffuse via the permeable membrane. The collagen layer is exposed to air to ensure oxygen supplies for the long-term preservation of organoids [150]. The latter approach is very promising and can also be applied for future glioblastoma research. In a recent study, the specific oncolytic activity of Zika virus against GSCs in glioblastoma cerebral organoids was demonstrated. The authors showed that SOX2 and integrin α_v_β_5_ represent key markers for Zika virus infection in association with suppression of immune response genes. Thus, Zika virus infection provides the possibility for brain tumor therapy [151]. The organoids established by Jacob et al. [117] are the first that, besides tumor cells, also include the TME. As CAR T cells represent a powerful new approach to treat glioblastoma, these glioblastoma organoids, which preserve the immune microenvironment and other stromal cells, were used as a model. The authors demonstrated that this rapid protocol for organoid generation provides a platform to test and optimize CAR T therapies for tumors of non-epithelial origin and enables a personalized treatment approach. We also showed that organoids established by this protocol after 4 weeks in culture included GSCs, differentiated glioblastoma cells, tumor vasculature, and immune cells, such as macrophages, microglia, and T cells (Figure 2, our results).

### 4.4. Animal Models

Syngeneic mouse models represent one of the oldest preclinical models for investigating antitumor therapies, in which spontaneous or chemically/virus-induced tumor cell lines from inbred mice are expanded in vitro and then inoculated into the same inbred mouse strain with an intact immune system [152]. The advantages of these models are their ease of use, rapid and reproducible expansion, and the possibility of genetic manipulation [120], especially to evaluate the efficacy of immunotherapeutic agents. However, these models, if implanted with tumor cell lines, lack genomic and microenvironmental heterogeneity due to the limited availability of CSCs that evolve genetic and epigenetic alterations that allow them to differentiate into multiple tumor cell types [153]. The GL261 syngeneic murine model represents one of the best characterized syngeneic, immunocompetent models in glioblastoma immunotherapy preclinical research [152]. Reardon et al. showed that blockade of CTLA-4, PD-1, or PD-L1 alone can eradicate glioblastoma growth in GL261 syngeneic murine models [154]. CAR T cells were shown to inhibit GL261/EGFRvIII tumor growth [155], and the potential of ErbB2-specific CAR-NK (NK-92/5.28) cells was demonstrated for adoptive immunotherapy of glioblastoma [100]. However, further studies are needed to determine whether these murine glioma models faithfully reflect human glioblastoma.

Several syngeneic rat glioma models are currently available for preclinical studies. However, rat glioma models, such as C6, showed immunological instability, since implanted tumor cells that should be syngeneic, triggered allogeneic immune response and lack of tumor growth because C6 glioma cells arose from an outbred strain of Winstar rat. Thus, these models are not useful for evaluating the efficacy of immunotherapy [156].

Genetically engineered mouse tumor models are generated through the introductions of genetic mutations specific to particular human cancers. Genetically engineered mouse tumor models of glioblastoma require gene expression manipulation using Tet regulation, Cre-inducible gene alleles [157], or the replication-competent avian leukosis virus splice-acceptor/avian tumor virus receptor A (RCAS/TVA) system, which uses retroviral or adenoviral vectors to deliver Cre recombinase for somatic cell gene transfers [158]. These models reflect the histology and biology of human glioblastoma; however, the differences in the TME and immune system between mice and humans reduce the clinical relevance of such cancer immunotherapy studies [121].

An alternative model system, patient-derived xenografts (PDXs), is also used in cancer research. PDX models of glioblastoma are based on subcutaneous or intracranial transplantation of patient-derived tumor cells, organoids, or tissues into immunodeficient NSG (NOD scid gamma) mice. This model better recapitulates the heterogeneity and complexity of the tumor and represents a valuable tool to investigate the characteristics of glioblastoma [110,122]. Furthermore, PDXs are commonly used to study the CAR T immunotherapeutic response [90] due to the lower chance of graft-versus-host rejection. One of the major limitations of these models is the need to use immunodeficient host strains for tumor engraftment and propagation. Because of the absence of functional elements of the immune system, such as NK cells, macrophages, and Tregs, the current PDX models are also unable to accurately assess the effects of different immunotherapies [90].

Humanized mice tumor models are generated by the engraftment of human tumor cell lines, CSCs, or human PDX tumors into immunodeficient NSG mice with an HLA-matched human immune system, which is initiated by the transplantation of human PBMCs, isolated from human adult blood, or CD34^+^ hematopoietic stem cells (HSCs). Transplanted CD34^+^ HSCs in immunocompromised mice differentiate into human helper T cells, cytotoxic T cells, B cells, monocytes, NK cells, and dendritic cells [123]; after tumor implantation, these mice can survive several months with a relatively stable percent of human cells in the blood. Human microglia/macrophage-like cells have also been developed in the brain of CD34^+^ HSC humanized mice [159]. This model is mostly used to evaluate treatment with anti-PD-1 and anti-CTLA-4 antibodies [160]. For example, in the study by Capasso et al. [161], nivolumab (anti-PD-1 antibody) inhibited MDA-MB-231 triple-negative breast cancer cells and CRC172 colorectal cancer cells in the humanized umbilical cord blood-derived HSC mouse models. Furthermore, the therapeutic antitumor potential of highly cytotoxic allogeneic super-charged NK cells was confirmed using an alternative humanized BLT (bone marrow, liver, thymus) mice model that was implanted with oral CSCs. The BLT model improves the functionality of T and NK cells via co-transplantation of fetal liver and thymus [113,162]. The main difficulty of HSC mouse models is their long-term establishment, and thus PBMCs from adult donors can be used to quickly restore the autologous human immune system [124]. However, the lifespan of PBMCs in mice is very short, i.e., only 3 weeks. As such, the timeframe to evaluate immunotherapies is reduced. These models are also likely to generate stable graft-versus-host reactions [163]. Moreover, the human CD45^+^ fraction in peripheral blood is composed mainly of T cells, limiting the investigation of other immune cells, such as monocytes and NK cells [110]. Different studies demonstrated that humanized mice with PBMCs can be successfully used for the evaluation of monoclonal antibodies, cytokine therapy (IL-2), immune checkpoint inhibitors, and dendritic cell-based vaccines [124,164,165]. For example, the efficacy of the anti-PD-1 antibody was evaluated using humanized NOG-dKO mice, in which human PBMCs and the glioblastoma cell line U87 were transplanted [166]. There are both advantages and disadvantages to this model; however, the humanized mouse platform is being improved in a way that the investigation of immunotherapeutics may become more predictive. Currently, the use of humanized mice models in glioblastoma preclinical and clinical studies is limited due to the lack of knowledge and remaining unanswered questions, including whether humanized mice models recapitulate the clinical features of glioblastoma patients.

## 5. Conclusions

We have summarized the recent findings on the progress of glioblastoma immunotherapy, the unique properties of glioblastoma that affect immunotherapy resistance, and tumor models that can facilitate our understanding of the fundamental immunobiology of glioblastoma and test potential novel immunotherapeutic approaches. Immunotherapy to fight glioblastoma holds great promise; however, there are many challenges, including (1) inter- and intra-tumor heterogeneity, (2) high immunosuppression in the TME, (3) a poor understanding of the mechanisms of immune cell activation in intracranial compartments, (4) the presence of tumor-initiating and therapy-refractory CSCs, and (5) the lack of appropriate tumor models to study combinatorial approaches with standard treatments and to predict treatment responses. Recent improvements in the establishment of glioblastoma organoids that exhibit tumor heterogeneity and include immune compartments as well as immuno-geno(pheno)typing of patient tumors hold great promise to help us resolve the complex immunobiology of brain tumors and to increase the efficiency of immunotherapy.

## Figures and Tables

**Figure 1 cells-10-00265-f001:**
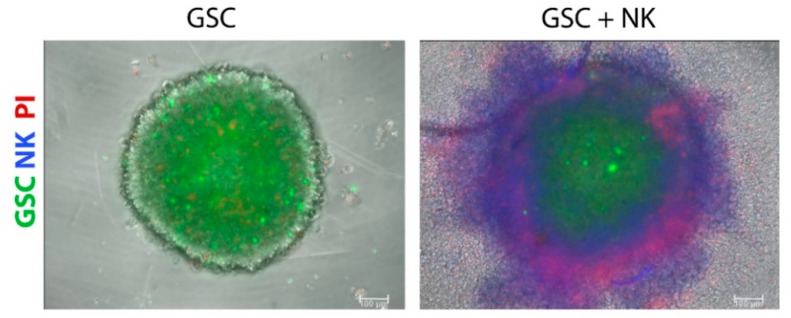
Super-charged natural killer (NK; blue) cell treatment decreased the number of glioblastoma stem cells (GSCs; green) and increased the number of dead cells (PI, red) in 3D tumorsphere models. NK cells were added to GSC tumorspheres at a NK:GSC ratio of 10:1, and images were acquired using an inverted fluorescence microscope 4 h later. Propidium Iodide (PI) staining was used to detect dead cells (red). Scale bars: 100 µm.

**Figure 2 cells-10-00265-f002:**
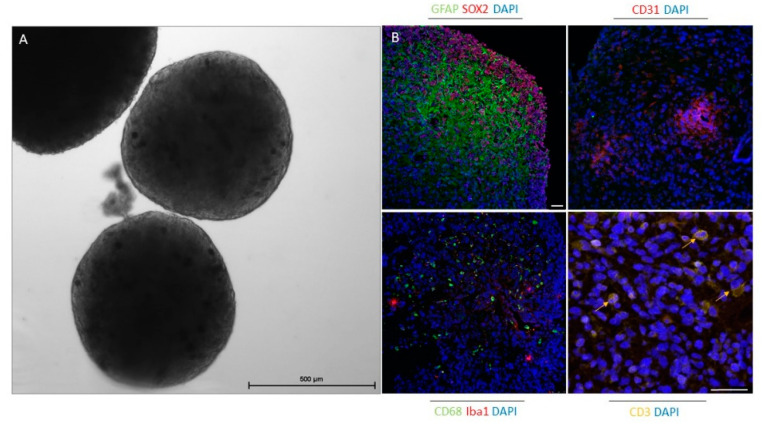
Glioblastoma organoids after 4 weeks in culture preserve specific elements of the tumor microenvironment. (**A**) Phase-contrast image of glioblastoma organoids in culture. Scale bar: 500 µm. (**B**) Immunofluorescence staining of paraffin-embedded glioblastoma organoids for glioblastoma stem cell marker (SOX2), differentiated glioblastoma cell and astrocyte marker (GFAP), endothelial cell marker (CD31), macrophage marker (CD68), microglia marker (Iba1), and T cell marker (CD3). Cell nuclei were stained with DAPl (blue). Scale bars: 50 µm.

**Table 1 cells-10-00265-t001:** A comparison of different glioblastoma tumor models for studying immunotherapy.

Tumor Model	Description	Advantages	Disadvantages	References
**In Vitro**				
Tumor cell lines	established tumor cell lines, grown as monolayers in serum-containing media	+ rapid expansion + low costs + long tradition + easy genetic manipulation + well-characterized + simple	- clonal selection in cell cultures based on media selection - lack of clonal diversity and heterogeneity - lack of TME and ECM	[111]
Cancer stem cells	patient-derived tumor cells grown in serum-free and growth factor-supplemented media as tumorspheres	+ reflect stem-like features and therapeutic resistance + preserve the tumor’s genetic background + phenotypic heterogeneity + 3D model	- lack of TME and ECM - clonal selection	[112]
Cell co-cultures	2D or 3D co-cultures of tumor and non-tumor cells, such as immune cells and stromal cells	+ heterotypic cellular interactions + simple + mechanistic studies of cellular cross-talk in TME	- lack of complex TME and architecture	[113]
Organotypic tissue slice cultures	precision-cut slices of tumor tissue, mounted onto porous membranes for mechanical support, and cultured in a controlled conditions	+ recapitulate TME + preserve inter-intra-tumoral heterogeneity and heterotypic cellular interactions + clinically relevant therapeutic response + platform for studying the tumor immune cell environment + tumor cell invasion model system	- limited by the availability of fresh patient samples - short lifespan - cryopreservation method is not optimized - not adapted for high throughput analysis	[114,115]
Patient-derived organoids	3D in vitro tissue constructs composed of multiple cell types, patient-based from resected tumors	+ preserve inter-intra-tumoral heterogeneity and heterotypic cellular interactions + preserve the tumor’s genetic background + recapitulate TME + pre-clinical applications + 3D model + high through-put + clinically relevant therapeutic response + feasibility of co-culture with immune cells	- variable ability to maintain over very long periods - limited by the availability of fresh patient samples - limited immune component - lack of model optimization - do not recapitulate tumor initiation	[116,117]
Genetically- engineered cerebral organoids	3D in vitro tissue constructs created by using genetic manipulations to induce tumorigenesis in cerebral organoids	+ 3D model + good reproducibility + clinically relevant therapeutic response + enable to study early phases of tumorigenesis and tumor progression + brain tissue architecture	- poorly recapitulate TME - the tumor’s genetic background is not preserved - lack of immune component	[118,119]
**In Vivo**				
Syngeneic mouse model	derived by transplanting mouse tumor cell lines or CSCs into strain-matched mice	+ immune system and response + present TME + simple with a long tradition + allows genetic modifications + tumor cell heterogeneity and clonal diversity with implanted CSCs	- limited tumor cell heterogeneity and clonal diversity with implanted tumor cell line - high costs - laborious, time-consuming - lack of human tumor-immune cell interactions - TME is of rodent origin	[110,120]
Genetically engineered mouse tumor model	created by introducing genetic modifications that result in spontaneous tumor development	+ allows genetic modifications + tumor cell heterogeneity and clonal diversity + tumor-immune cell interactions if immunocompetent mice are used + present TME	- large number of animals - laborious, time-consuming - poor inter-animal comparability - high costs - TME is of rodent origin	[121]
Patient-derived xenografts	derived by transplanting human tumor explants into immunodeficient mice	+ tumor cell heterogeneity and clonal diversity + present TME + reflect tumors in human + little graft-versus-host rejection for adoptive cell therapy (CART) + preserve the tumor’s genetic background	- high costs - fail to develop a functional immune system - lack of human tumor-immune cell interactions - laborious, time-consuming - TME is of rodent origin	[90,122]
Humanized mouse tumor model	generated by the engraftment of human cancer cell lines or human PDX tumors into mice with a reconstituted human immune response	+ tumor heterogeneity and clonal diversity + present TME + human immune cells + mimicking human tumor and immune system interactions + realistic representation of immunotherapy safety and clinical response + preserves the tumor’s genetic background	- long-lasting establishment - high costs - laborious, time-consuming - slow tumor growth	[110,123,124]

CSC: cancer stem cell; ECM: extracellular matrix; PDX: patient-derived xenografts; TME: tumor microenvironment.

## Data Availability

No datasets were generated during the current study.

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
