# Peer review of "Immunotherapy of Glioblastoma: Current Strategies and Challenges in Tumor Model Development"

_cells, 2021, doi:10.3390/cells10020265_

Round 1

Reviewer 1 Report

The manuscript “Immunotherapy of glioblastoma: current strategies and challenges in tumor model development” by Bernarda Majc and colleagues represents a thorough review of recently developed immunotherapies for glioblastoma and provides an in-depth summary of pre-clinical in vitro and in vivo models for studying glioblastoma biology and testing therapeutic strategies. The manuscript is timely, well-organized, and concise. The authors do a great job balancing the coverage of different immunotherapeutic approaches, pre-clinical model systems, and their advantages and disadvantages. My major suggestion for improving this manuscript is to make a more obvious distinction between patient-derived glioblastoma organoids vs. genetically engineered cerebral organoids since they do not share many characteristics and have distinct advantages over each other depending on the goal of the study (see comments). Otherwise the manuscript was an enjoyable and informative read that will benefit the field.

Comments:

  1. I think it is important to make a more obvious distinction between patient-derived glioblastoma organoids vs. genetically altered cerebral organoids similar to how you distinguish “genetically engineered mouse tumor model” and “patient-derived xenografts. It would be better to separate these two organoid models in Table 1 since they do not share many of the same advantages and disadvantages. For example, patient-derived organoids do not recapitulate tumor initiation very well since you are capturing the tissue far after initiation, but patient-derived organoids better recapitulate the cellular heterogeneity and characteristics of individual patients’ tumors since they are derived directly from the tissue. Genetically engineered cerebral organoids do not preserve the entirety of the tumor’s genetic background and poorly recapitulate the TME.
  2. It would be worth mentioning, at least briefly, recent efforts to use Zika Virus’s tropism for neural stem cells as an oncolytic virus for glioblastoma therapy. Jeremy Rich’s group has a few publications (https://doi.org/10.1016/j.stem.2019.11.016).
  3. Consider adding citation (https://doi.org/10.1038/s41596-020-0402-9) to glioblastoma organoids.
  4. Consider adding a description of organotypic slice cultures for glioblastoma modeling (https://doi.org/10.1093/neuonc/not003 and https://doi.org/10.3791/53557) to Table 1 and/or section 4 since this system is valuable for modeling tumor migration and potential interactions with cellular immunotherapies.
  5. Table 1 would be easier to read if the text in the columns was left justified instead of centered.

Author Response

Dear Reviewer,

We thank you for all valuable comments and suggestions. All amendments in the manuscript are in track changes. We responded step by step to the reviewers’ comments in the following paragraphs.

Reviewer 1

The manuscript “Immunotherapy of glioblastoma: current strategies and challenges in tumor model development” by Bernarda Majc and colleagues represents a thorough review of recently developed immunotherapies for glioblastoma and provides an in-depth summary of pre-clinical in vitro and in vivo models for studying glioblastoma biology and testing therapeutic strategies. The manuscript is timely, well-organized, and concise. The authors do a great job balancing the coverage of different immunotherapeutic approaches, pre-clinical model systems, and their advantages and disadvantages. My major suggestion for improving this manuscript is to make a more obvious distinction between patient-derived glioblastoma organoids vs. genetically engineered cerebral organoids since they do not share many characteristics and have distinct advantages over each other depending on the goal of the study (see comments). Otherwise the manuscript was an enjoyable and informative read that will benefit the field.

Comments:

  1. I think it is important to make a more obvious distinction between patient-derived glioblastoma organoids vs. genetically altered cerebral organoids similar to how you distinguish “genetically engineered mouse tumor model” and “patient-derived xenografts. It would be better to separate these two organoid models in Table 1 since they do not share many of the same advantages and disadvantages. For example, patient-derived organoids do not recapitulate tumor initiation very well since you are capturing the tissue far after initiation, but patient-derived organoids better recapitulate the cellular heterogeneity and characteristics of individual patients’ tumors since they are derived directly from the tissue. Genetically engineered cerebral organoids do not preserve the entirety of the tumor’s genetic background and poorly recapitulate the TME.

Reply: We agree with the reviewer and we made adjustments as suggested. We separated two organoid models in the Table 1 into patient-derived organoids and genetically-engineered cerebral organoids and added descriptions of advantages and disadvantages of each model (please see Table 1).

  1. It would be worth mentioning, at least briefly, recent efforts to use Zika Virus’s tropism for neural stem cells as an oncolytic virus for glioblastoma therapy. Jeremy Rich’s group has a few publications (https://doi.org/10.1016/j.stem.2019.11.016).

Reply: We added the suggested description of Zika Virus-specific targeting of GSCs in organoid model from publication of Zhu and co-workers: “In a recent study, specific oncolytic activity of Zika virus against GSCs in glioblastoma cerebral organoids was demonstrated. The authors showed that SOX2 and integrin αvβ5 represent key markers for Zika virus infection in association with suppression of immune response genes. Thus, Zika virus infection provides the possibility for brain tumor therapy [152]Please see lines 516-520.

  1. Consider adding citation (https://doi.org/10.1038/s41596-020-0402-9) to glioblastoma organoids.

Reply: The reference was added to the organoid section (Section 4) in the text. See the line 483 and reference 147.

We also added description of a relevant recent publication by Linkous and co-authors, who developed glioma cerebral organoid model with hESC-derived cerebral organoids and patient-derived GSCs to recapitulate cellular behavior of glioblastoma and to maintain genetic aberrations found in the original tumor. Please see lines 479-483:

“A novel approach using hESC-derived cerebral organoids and patient-derived GSCs to model tumor cell invasion was recently developed, i.e., a glioma cerebral organoid mod-el. This system was shown to recapitulate cellular behavior of glioblastoma and to maintain genetic aberrations found in the original tumor [144].”

  1. Consider adding a description of organotypic slice cultures for glioblastoma modeling (https://doi.org/10.1093/neuonc/not003 and https://doi.org/10.3791/53557) to Table 1 and/or section 4 since this system is valuable for modeling tumor migration and potential interactions with cellular immunotherapies.

Reply: We than reviewer for this comment. The description of organotypic slice cultures for glioblastoma modeling was added to the Table 1 and to the text. Please see lines 430-442:

“4.2. Organotypic tissue slices

Organotypic tissue slice model of glioblastoma represent precision-cut slices of tumor tissue, in which the original inter and intra-tumor heterogeneity and the architecture of the tumor is maintained. Slices of tumor are prepared with an automated vibratome and transferred onto membrane culture inserts for mechanical support in specific cultivation medium [115]. This technique is relatively fast, it does not involve selective out-growth of tumor cells, and therefore can be used for personalized treatment. Organotypic cultures have been used to study the invasive properties of glioblastoma and patient-specific effect of anti-invasive drugs [116]. Recently an organotypic slice culture technique was developed from fresh pancreatic ductal adenocarcinoma to study the immune response after immunotherapy treatment [133], and can be applied to a variety of solid tumors, including glioblastoma. A disadvantage of this models is its relatively low through-put. The technique is laborious and requires specialized analysis tools.”

Moreover, the suggested references were added to the text (115 and 116).

  1. Table 1 would be easier to read if the text in the columns was left justified instead of centered.

Reply: We corrected the Table 1.

Reviewer 2 Report

The authors did a great job summarizing the current state of IT in GBM (the modes to do so) and the models, both in vitro and in vivo, to study tumor – immune cell interactions.

Some minor comments:

  1. The authors claim a high infiltration of neutrophils in brain, but I’m not sure this is the case. The paper the authors refer to only observe a gene signature of neutrophils, but based on our experience we hardly see any neutrophils in GBM, regardless of the subtype
  2. Regarding the discussion on TMB, the authors should adapt their text and discuss this paper: https://www.nature.com/articles/s41467-020-20469-6
  3. Regarding CSF1R – the authors should state that clinical trials (both ND and Rec setting) with CSF1R inhibitors were negative, in addition to some resistance mechanisms that were described
  4. The authors should add some statements on the defective mobilization of T cells from the periphery in GBM patients. This obviously can significantly hinder IT approaches.
  5. Regarding immune-checkpoint inhibitors; the authors should also discuss the timing of delivery (e.g. neoadjuvant vs adjuvant treatment)
  6. The table is not very easy to interpret – perhaps the layout can be adapted to make it more accessible

Author Response

Dear Reviewer,

We thank you for all valuable comments and suggestions. All amendments in the manuscript are in track changes. We responded step by step to the reviewers’ comments in the following paragraphs.

Reviewer 2

The authors did a great job summarizing the current state of IT in GBM (the modes to do so) and the models, both in vitro and in vivo, to study tumor – immune cell interactions.

Some minor comments:

  1. The authors claim a high infiltration of neutrophils in brain, but I’m not sure this is the case. The paper the authors refer to only observe a gene signature of neutrophils, but based on our experience we hardly see any neutrophils in GBM, regardless of the subtype

Reply: Thank you for this comment. We agree with the reviewer, therefore we have changed the sentence in lines 69-70, now written: “Mesenchymal tumors contain abundant gene expression signature for macrophages, CD4+ T cells, and neutrophils [22]; this is also associated with a higher glioma grade [19].”

  1. Regarding the discussion on TMB, the authors should adapt their text and discuss this paper: https://www.nature.com/articles/s41467-020-20469-6

Reply: We thank reviewers for this valuable comment. We have corrected and expanded our part on mutational burden and included this reference, now written in lines 358-368:

“It has been shown that radiation increases the mutational burden of tumors and triggers tumor necrosis and antigen release, leading to increased antigen presentation and immunogenicity [107]. High mutational burden is associated with response to immunotherapy in several types of cancer, but not in gliomas. For example, gliomas with high mutational burden and mismatch repair genes deficiency are less responsive to PD-1 blockage [108]. Low mutation burden in recurrent glioblastoma patients was recently associated with longer survival after immunotherapy, implicating that tumor mutational burden itself may not be a causative driver of response to immunotherapy, but may reflect immunological status of tumor or some other co-related feature, among them time to recurrence, TP53 mutation and any differences in the clinical care between patients with high vs. low mutational burden [109].”

  1. Regarding CSF1R – the authors should state that clinical trials (both ND and Rec setting) with CSF1R inhibitors were negative, in addition to some resistance mechanisms that were described

Reply: We added to the text description of results of clinical trials with CSF-1R inhibitors. Please see the lines 342-347:

“Although CSF-1R inhibitors showed promising results in preclinical studies, the clinical trials with orally administered CSF-1R inhibitor PLX-3397 were negative, with minimal clinical efficacy in patients with recurrent glioblastoma. Microenvironment-driven resistance to CSF-1R inhibitor is mediated through phosphatidylinositol 3-kinase (PI3K) pathway, that was elevated and driven by insulin-like growth factor–1 (IGF-1) and tumor cell IGF-1 receptor (IGF-1R) [103, 104].”

  1. The authors should add some statements on the defective mobilization of T cells from the periphery in GBM patients. This obviously can significantly hinder IT approaches.

Reply: We thank the reviewer for pointing out the defective mobilization of T cells from periphery of glioblastoma patients. We provided main mechanism of defective T cell mobilization that is most commonly listed in literature and is mediated by loss of the surface sphingosine-1-phosphate receptor 1 (S1P1) and entrapment of T cells in bone marrow in lines 105-111. Please see:

“Moreover, defects in antitumor responses arise from defective T cell mobilization from periphery due to T cell entrapment in the bone marrow, which is caused by the loss of the surface sphingosine-1-phosphate receptor 1 (S1P1) [42,43] that binds the lipid second messenger sphingosine-1-phosphate (S1P) [44]. The S1P-S1P1 axis plays a role in governing lymphocyte trafficking. Naïve T cell egress from bone marrow or secondary lymphoid organs cannot occur without functional S1P1 on the cell surface, as S1P1 is essential for lymphocyte recirculation [42, 45].”

  1. Regarding immune-checkpoint inhibitors; the authors should also discuss the timing of delivery (e.g. neoadjuvant vs adjuvant treatment)

Reply: We thank the reviewer for that suggestion. We have included comparison of the immune-checkpoint inhibitor timing treatment, neoadjuvant vs. adjuvant, in the text and added manuscript by Cloughesy et al. (Nat. Med. 2019, 89). Please see lines 260-261, 269-274:

“Several reasons for the poor efficacy of immune checkpoint inhibitors in glioblastoma have been identified, including the timing of delivery (neoadjuvant or adjuvant therapy), BBB, low infiltration of T cells into the tumor, predominant myeloid infiltrate, and multi-layered immunosuppression in the TME [84–88].” AND

“Recent study by Cloughesy et al. have shown that patients with recurrent glioblastoma that received neoadjuvant treatment with pembrolizumab (anti-PD-1), with continued adjuvant therapy following surgery, had significantly improved overall survival compared to that receiving only adjuvant postsurgical treatment with pembrolizumab. Neoadjuvant administration of pembrolizumab enhanced local and systemic immune responses in patients [89].”

  1. The table is not very easy to interpret – perhaps the layout can be adapted to make it more accessible

Reply: We corrected the Table 1.

Reviewer 3 Report

Review cells-1069051

Title: “Immunotherapy of glioblastoma: current strategies and challenges in tumor model development”

The authors present an important review of the current evidence regarding immunotherapeutic approaches for the treatment of glioblastoma. Evidently, this is clinically and scientifically very sensitive topic after all clinical trials based on this principle have reported negative results in GBM patients. The issue highlighted in this review is the discrepancy between the partly promising data derived from preclinical research and the rather sobering results of clinical studies. Although this is without doubt an interesting paper, there are a few aspects, which need to be addressed before being published in Cells.

  1. Abstract: “Glioblastoma remains one of the most common and aggressive brain malignancies” is a very vague wording. Pls. change to “GBM is the most common primary malignant brain tumor in the adult population…”
  2. Introduction: GBM is not listed as astrocytoma grade IV anymore in the revised version of the WHO classification. Pls. adjust accordingly.
  3. The sentence “novel treatment modalities…” need to include tumor treating fields, which have shown to be effective in GBM in a phase III trial.
  4. Why is the “intracranial localization” responsible for poor treatment response in GBM?
  5. The statement “the BBB is disrupted” is not adequate. The BBB alteration in GBM is highly variable and vastly heterogeneous. Pls. adjust and provide references.
  6. Chapter 2 contains very important information regarding the immunosuppressive tumor environment in GBM, however it is really hard to read since it lacks structuring. It would be desirable, if the authors could summarize the contents thematically under specific main aspects, instead of reporting facts and references in a discursive fashion.
  7. When discussing the vaccination against EGFRvIII in GBM, it is necessary to report the complete futility of this approach in a multinational phase III trial (Weller et al. Lancet Oncology 2017).
  8. To complete the chapter about immuncheckpoint inhibition, it would be necessary to include the study reported by Cloughesy et al., which indicates a positive survival signal after neoadjuvant pembrolizumab, with continued adjuvant therapy following surgery.
  9. In the animal model chapter, it would be helpful to address the fact that implanted rat glioma models are inherently immunologically instable, since the implanted tumor cells, despite being “syngeneic” are immunogenic. Immunotherapeutic approaches as well as gene therapy experiments have shown curative effects, which are rather based on a transplant rejection rather than true anti – tumor effects.

Author Response

Dear Reviewer,

We thank you for all valuable comments and suggestions. All amendments in the manuscript are in track changes. We responded step by step to the reviewers’ comments in the following paragraphs.

Reviewer 3

Title: “Immunotherapy of glioblastoma: current strategies and challenges in tumor model development”

The authors present an important review of the current evidence regarding immunotherapeutic approaches for the treatment of glioblastoma. Evidently, this is clinically and scientifically very sensitive topic after all clinical trials based on this principle have reported negative results in GBM patients. The issue highlighted in this review is the discrepancy between the partly promising data derived from preclinical research and the rather sobering results of clinical studies. Although this is without doubt an interesting paper, there are a few aspects, which need to be addressed before being published in Cells.

1.Abstract: “Glioblastoma remains one of the most common and aggressive brain malignancies” is a very vague wording. Pls. change to “GBM is the most common primary malignant brain tumor in the adult population…”

Reply: We changed this sentence as suggested by the reviewer (lines 14-15).

2. Introduction: GBM is not listed as astrocytoma grade IV anymore in the revised version of the WHO classification. Pls. adjust accordingly.

Reply: We thank reviewer to point this out. We adjusted the text and changed astrocytoma to glioma grade IV (line 36).

3. The sentence “novel treatment modalities…” need to include tumor treating fields, which have shown to be effective in GBM in a phase III trial.

Reply: The tumor treating fields treatment was added to the text, lines 38-40:

»For example, tumor treating fields treatment together with chemotherapy improved median overall survival of glioblastoma patients from 16 to 20.9 months [2].«

4. Why is the “intracranial localization” responsible for poor treatment response in GBM?

Reply: We realized that we wanted to point out here the highly invasive nature of GBM and GBM tumor cells, which invade the brain parenchyma, few centimeters from tumor bulk and are hard to detect and be surgically removed. We corrected this sentence, lines 44-46:

“The poor response of glioblastoma to treatment and its poor prognosis are associated with diffused invasion pattern within the central nervous system (CNS) [7].

5. The statement “the BBB is disrupted” is not adequate. The BBB alteration in GBM is highly variable and vastly heterogeneous. Pls. adjust and provide references.

Reply: We agree with the reviewer. We corrected the text and added the sentence on variable BBB disruption: “In addition, the BBB can be disrupted in brain tumor patients, which increases infiltration of immune cells into the tumor area. However, most of GBM patients have variable regions of disrupted BBB, meaning that tumor regions with disrupted BBB and tumor regions with intact BBB exist [14].” Please see the lines 54-57.

6. Chapter 2 contains very important information regarding the immunosuppressive tumor environment in GBM, however it is really hard to read since it lacks structuring. It would be desirable, if the authors could summarize the contents thematically under specific main aspects, instead of reporting facts and references in a discursive fashion.

Reply: We restructured this chapter based on several aspects of glioblastoma immunosuppression. We start with description of systemic immunosuppression and then continue with local immunosuppression in glioblastoma microenvironment. The reasons of local immunosuppression are then briefly described and elaborated in each paragraph: mechanisms of glioblastoma cell and GSC immune-evasion, description of low NK and T cell infiltration, mechanisms of T and NK cell dysfunction, mechanisms of action of specific tumor immunosuppressive factors and immunosuppressive cells. Please see lines 96-180.

7. When discussing the vaccination against EGFRvIII in GBM, it is necessary to report the complete futility of this approach in a multinational phase III trial (Weller et al. Lancet Oncology 2017).

Reply: This suggested study was added to the text, lines 197-200:

“Moreover, a randomized, double-blind and international phase 3 trial, which assessed the efficacy of vaccine, based on EGFRvIII-specific peptide (CDX-110), with temozolomide did not show a survival benefit for newly diagnosed glioblastoma patients with EGFRvIII mutation [74].”

8. To complete the chapter about immuncheckpoint inhibition, it would be necessary to include the study reported by Cloughesy et al., which indicates a positive survival signal after neoadjuvant pembrolizumab, with continued adjuvant therapy following surgery.

Reply: We thank the reviewer to point out this very interesting study. The study reported by Cloughesy et al. was added to the text, line 269-274:

“Recent study by Cloughesy et al. have shown that patients with reccurent glioblastoma that received neoadjuvant treatment with pembrolizumab (anti-PD-1) with continued adjuvant therapy following surgery, had significantly improved overall survival compared to that receiving only adjuvant post-surgical treatment with pembrolizumab. Neo-adjuvant administration of pembrolizumab enhanced local and systemic immune responses in patients [89].”

9. In the animal model chapter, it would be helpful to address the fact that implanted rat glioma models are inherently immunologically instable, since the implanted tumor cells, despite being “syngeneic” are immunogenic. Immunotherapeutic approaches as well as gene therapy experiments have shown curative effects, which are rather based on a transplant rejection rather than true anti – tumor effects.

Reply: We thank reviewer for this valuable comment. We added the text on rat glioma models in lines 555-560:

“There are also several syngeneic rat glioma models that are currently available for preclinical studies. However, rat glioma models, such as C6, showed immunological instability, since implanted tumor cells that should be syngeneic, triggered allogeneic immune response and lack of tumor growth, because C6 glioma cells arose from outbred strain of Winstar rat. Thus, these models are not useful for evaluating the efficacy of immunotherapy [158].”

Round 2

Reviewer 3 Report

The authors have carefully adressed all aspects raised in my review. ONly the term "glioma WHO grade IV" needs to be changed into "Glioblastoma WHO grade IV". The paper is now suitable for publication, and does not not have to be reviewed by me again.